# Design of Miniaturized and Wideband Four-Port MIMO Antenna Pair for WiFi

**DOI:** 10.3390/mi15070850

**Published:** 2024-06-29

**Authors:** Yao Hu, Yongshun Wang, Lijun Zhang, Mengmeng Li

**Affiliations:** 1School of Electronic and Information Engineering, Lanzhou Jiaotong University, Lanzhou 730070, China; wangysh@mail.lzjtu.cn (Y.W.); zhanglj_81@tsnu.edu.cn (L.Z.); ll206063147@163.com (M.L.); 2School of Mechanical Engineering, Lanzhou Jiaotong University, Lanzhou 730070, China; 3School of Electronic Information and Electrical Engineering, Tianshui Normal University, Tianshui 741000, China

**Keywords:** multiple-input multiple-output (MIMO) antenna, mobile terminal, orthogonal, Wi-Fi, wideband, low coupling

## Abstract

A miniaturized and wideband four-port multiple-input multiple-output (MIMO) antenna pair for Wi-Fi mobile terminals application is proposed. The proposed antenna pair consists of four multi-branch antenna elements arranged orthogonally, with an overall size of 40 × 40 × 3.5 mm^3^ and each antenna element size of 15.2 × 3.5 mm × 0.8 mm^3^. The performance of the proposed antenna shows the advantages of a wide frequency band, low mutual coupling, high efficiency, and a compact structure. The wideband characteristics of the antenna elements are achieved through multi-mode resonance. The suppression of coupling is accomplished by strategically positioning the four compact antenna elements to ensure their maximum radiation directions are orthogonal, thus eliminating the need for an additional decoupling structure. In this paper, the proposed antenna is optimized in terms of the parameters then simulated and measured. The simulated results illustrate that an impedance bandwidth of the antenna is about 15% (5.06~5.88 GHz) with S_11_ < −10 dB, excellent port isolation exceeds 20 dB between all ports, a high radiation efficiency ranges from 51.2% to 89.9%, the maximum gain is 4.5 dBi, and the ECCs are less than 0.04. The measured results show that the −10 dB impedance bandwidth of the antenna is about 13% (5.13~5.80 GHz), the isolation between the antenna elements is better than 21 dB, the radiation efficiency ranges from 51.8% to 92.3%, the maximum gain is 5.3 dBi, and the ECCs are less than 0.05. The proposed four-port MIMO antenna works on the 5G LTE band 46 and Wi-Fi 6E operating bands. As a mobile terminal antenna, the proposed design scheme demonstrates excellent performance and applicability, fulfilling the requirements for 5G mobile terminal applications.

## 1. Introduction

With the rapid development of 5G mobile wireless communication technology, users have higher requirements for higher data transmission rates, data throughput increase, and data transmission reliability. Multiple-input multiple-output (MIMO) antenna technology employs multiple antennas at both the transmitter and receiver to achieve spatial multiplexing. This technology significantly enhances the channel capacity and data transmission reliability within a limited bandwidth, making it widely applicable in 5G wireless communication systems. In recent years, MIMO antennas, as a pivotal technology for 5G mobile communications, have emerged as a focal point of research in the fields of electronics and mobile communications. For MIMO antennas, due to the limitations of antenna size, spacing, and array size, mutual coupling inevitably results between the antennas, which leads to reduced antenna isolation, increased spatial correlation of radiation patterns, and a deteriorated signal-to-noise ratio (SNR) and channel capacity. Therefore, in recent years, a series of MIMO antenna decoupling techniques have been researched to improve the isolation and channel capacity of MIMO antennas. MIMO antenna decoupling techniques are mainly divided into two categories: external decoupling techniques and internal decoupling techniques. External decoupling techniques introduce additional structures between adjacent antenna elements to reduce mutual coupling, including neutralization lines (NL) [1,2,3,4], parasitic elements (PE) [5,6,7,8], defected ground structures (DGS) [9,10,11,12,13], metasurfaces [14,15,16], and electromagnetic band-gap (EBG) structures [17,18]. Among these designs, a neutralization line (NL) is positioned between two printed dual-monopole antenna systems to achieve excellent isolation between ports, with isolation better than −19 dB in [2]. In [6], parasitic meander lines are located between a printed, folded monopole antenna and a parasitic inverted-L element in a MIMO antenna system, achieving ultra-wideband decoupling. In [11], a defected ground structure with an enhanced I-shaped slot is employed to suppress coupling between MIMO circularly polarized antennas, achieving high isolation exceeding 35 dB within the −10 dB impedance bandwidth of 5.2–5.33 GHz. In [16], a metasurface-based decoupling method (MDM) consisting of pairs of non-uniform cut wires with two different lengths is proposed. This method achieves dual-band decoupling with isolation exceeding 25 dB at two independent bands of two coupled MIMO antennas. In [18], a mushroom-like Electromagnetic Band Gap (EBG) structure is implemented in the design of microstrip antenna arrays to suppress the strong surface wave coupling, resulting in an 8 dB mutual coupling reduction at the resonant frequency. However, all these external decoupling techniques need additional space for the implementation of decoupling structures to reduce mutual coupling, resulting in larger antenna sizes and complex antenna structures. Internal decoupling techniques include pattern diversity [19,20], spatial diversity [21], and polarization diversity [22,23,24]. In [21], the orthogonal mode technique is employed in an eight-element MIMO antenna to achieve excellent isolation without any decoupling structure at the operating frequency range of 3.4–3.6 GHz. The internal decoupling techniques resolve the mutual coupling between adjacent antenna elements by leveraging the inherent properties of the antennas, without adding any extra structures.

To meet the demands for high transmission rates and large capacity, MIMO antennas in mobile terminal applications such as smartphones, smartwatches, personal computers, and wireless routers are developing towards the trend of miniaturization, broadband, multi-frequency, and low coupling. In this study, we propose a miniaturized broadband four-port MIMO antenna for Wi-Fi applications by combining the radiation mechanism of the antenna and the internal decoupling technique of radiation direction diversity. Four identical antenna elements are orthogonally arranged to ensure independent signal radiation. This configuration avoids adding complexity in the antenna structure and enhances spatial efficiency. This design meets the requirements for miniaturization, wide bandwidth, and high isolation. The characteristics and application scenarios of the MIMO antenna were simulated and analyzed by HFSS 2022 version (High-Frequency Structure Simulator) electromagnetic simulation software. The simulation results indicate that the proposed 4 × 4 MIMO antenna achieves a −10 dB impedance bandwidth of approximately 15% (5.06–5.88 GHz), with isolation exceeding 20 dB among elements, and the envelope correlation coefficients (ECCs) are less than 0.04. Meanwhile, the S-parameters and radiation performance of the antenna were measured using a ZVA67 vector network analyzer and a microwave millimeter-wave antenna darkroom testing and analysis calibration system. The measurement results show that a −10 dB impedance bandwidth is approximately 13% (5.13–5.80 GHz), the element isolation is higher than 21 dB, the radiation efficiency is 51.8% to 92.3%, the maximum gain is 5.3 dBi, and the ECCs are less than 0.3. These simulation and test results confirm that the radiation performance of the proposed 4 × 4 MIMO Wi-Fi antenna satisfies the operational demands of 5G mobile devices.

## 2. Design of Proposed 4 × 4 MIMO (Multiple-Input Multiple-Output) Wi-Fi Antenna

### 2.1. Construction of the Proposed Antenna

The configuration of the proposed 4 × 4 MIMO antenna is shown in Figure 1, which is composed of two primary sections: a horizontal metal ground plane and vertical radiating elements. The ground plane (size: 40 × 40 mm^2^) is etched onto the reverse side of a 40 × 40 × 0.8 mm^3^ FR4 substrate with a relative permittivity of 4.4, relative permeability of 1, and loss tangent of 0.02. The four antenna radiating elements are printed on the vertical frame (size: 40 × 3.5 × 0.8 mm^3^), in which an L-shaped branch (size: 7.5 × 2.3 mm^2^) is printed on the inner side of the vertical frame and directly connected to the feed microstrip lines on the horizontal plane, and a radiating patch with a multi-branch structure (size: 15.2 × 3.5 mm^2^) is printed on the outer side of the vertical frame and directly connected to the ground plane. To achieve directional diversity, four adjacent antenna elements are arranged orthogonally. The port of each antenna element is connected to the back of the system substrate by means of 50 Ω SMA connectors via holes. The optimized antenna dimensions are specified in Table 1.

### 2.2. Evolution and Analysis of the Proposed Antenna

To elucidate the evolution of the antenna structure and comprehend the radiation mechanism, this section focuses on analyzing the evolution of antenna element 1’s structure and the simulated reflection coefficient results when port 1 is excited. The broadband design of the antenna is achieved by constructing multi-branch radiating patches of different sizes on the same plane, thereby exciting multiple resonant modes. Additionally, the RF signal is coupled with the external radiating patch through the internal patch by employing the microstrip antenna RF feeding, which also leads to the potential of broadband since this approach can increase the equivalent capacitance. Based on the above design ideas, the evolution of the antenna elements occurs through three models, and the proposed antenna finally achieves the characteristics of miniaturization, broadband, and low coupling. The evolution process of the proposed antenna element structure is shown in Figure 2.

The MIMO antenna can be regarded as a multi-port microwave network composed of multiple antennas. The energy transmission of the MIMO antenna is generally described by scattering parameters (S-parameters). This parameter can be used to determine the power transmission and reflection characteristics by analyzing the relationship between the normalized incident wave and reflected wave at each port, so as to obtain the impedance matching result and the isolation between antenna elements. Assuming the MIMO antenna system is an n-port network composed of n antenna elements, the incident wave powers are *a*_1_, *a*_2_, …, *a_n_*, and the reflected wave powers are *b*_1_, *b*_2_, …, *b_n_* in this network. The relationship between the incident wave and the reflected wave can be expressed by Equation (1).
(1)b=Sa,

This relationship can be expressed as Equation (2) by matrix form.
(2)$$\left[\begin{array}{c} b_{1}\\ b_{2}\\ \cdots \\ b_{n} \end{array}\right]=\left[\begin{array}{cccc} S_{11} & S_{12} & \cdots & S_{1n}\\ S_{21} & S_{22} & \cdots & S_{2n}\\ \cdots & \cdots & \cdots & \cdots \\ S_{n1} & S_{n2} & \cdots & S_{nn} \end{array}\right]\left[\begin{array}{c} a_{1}\\ a_{2}\\ \cdots \\ a_{n} \end{array}\right],$$
(3)$$S=\left[\begin{array}{cccc} S_{11} & S_{12} & \cdots & S_{1n}\\ S_{21} & S_{22} & \cdots & S_{2n}\\ \cdots & \cdots & \cdots & \cdots \\ S_{n1} & S_{n2} & \cdots & S_{nn} \end{array}\right],$$

Equation (3) represents the S-parameter matrix, also known as the scattering parameter matrix. In this matrix, *S_ii_* denotes the reflection coefficient, which indicates the ratio of the reflected wave to the incident wave at port *i* when port *i* is excited and other ports are terminated with matched loads. A smaller *S_ii_* value indicates better impedance matching of the antenna, and the value of *S_ii_* is generally required to be less than −6 dB or −10 dB. *S_ji_* represents the transmission coefficient, indicating the coupling degree between port *i* and port *j* when port *i* is excited and other ports are terminated with matched loads. *S_ji_* is also the index of port isolation, with a smaller value indicating higher isolation between ports, and the value of *S_ji_* is generally required to be less than −10 dB or −15 dB. The variation in reflection coefficients for the three evolutionary models of the designed antenna element is shown in Figure 3.

Since the radiating patch on the inner side of the vertical dielectric substrate is directly connected to the ground plane, altering its structure significantly impacts impedance matching. To extend the operational bandwidth, radiating patches of different lengths are constructed on the outer side of the vertical dielectric substrate, thereby creating different current paths. Initially, a rectangular loop structure comprising two inverted L-shaped coupling branches is constructed. The inner L-shaped branch feeds the outer rectangular loop structure through substrate coupling, exciting only one resonant mode at 5.9 GHz. However, the return loss is less than 10 dB, reflecting inadequate impedance matching and reduced radiative efficiency. In order to improve the impedance matching, Model 2 is designed to resonate at the 5.5 GHz by adding an inverted L-shaped branch to the outer side of the rectangular loop, which operates a frequency band of 5.33–5.68 GHz (*S*_11_ < −10 dB) and has a bandwidth of approximately 6%. To create more additional resonant modes and further increase the impedance bandwidth, Model 3 attempts to add an I-shaped coupling branch within the rectangular loop of the primary radiating element. This structure introduces resonant points at 5.2 GHz and 5.6 GHz, thus exciting two resonant modes and thus increasing the bandwidth of the antenna.

The simulated transmission coefficients of the proposed 4 × 4 MIMO antenna by HFSS are shown in Figure 4. The orthogonally arranged antenna pairs achieve radiation diversity, thereby reducing mutual coupling between antenna elements and enhancing channel isolation without using any decoupling structures. It can be observed that within the operating band of 5.13~5.80 GHz (*S*_11_ < −10 dB), the isolation of *S*_21_ is superior to 20.25 dB, *S*_31_ is better than 25 dB, and *S*_41_ is better than 20.11 dB, ensuring excellent isolation performance between adjacent antennas.

To elucidate the operational principle of the proposed 4 × 4 MIMO antenna, the current models and the current distribution of one antenna element are analyzed. Figure 5 shows the simulated current models of the proposed MIMO antenna at 5.2 GHz and 5.6 GHz. The surface current model is related to the antenna’s resonant behavior, directly affecting the operating state.

As shown in Figure 5a, when port 1 is excited and the other ports are terminated with a 50 Ω load at 5.2 GHz, the current exhibits a single mode and is mainly distributed at the two long edges of the outer rectangular loop and the I-shaped branch of the radiating element of antenna element 1. When port 1 operates at 5.6 GHz, the resonance mode changes. As shown in Figure 5b, the current distribution for this mode is primarily concentrated along the long edge of the outer rectangular loop and the I-shaped branch of the radiating element. The blue circle in the figure represents the current zero point. It can be observed that the current direction is opposite on either side of the zero point, with two maximum values. Therefore, this resonance mode corresponds to a half-wave mode. Consequently, the proposed 4 × 4 MIMO antenna achieves broadband performance through this multi-mode resonance coupling.

Figure 6 shows the simulated current distributions of the proposed 4 × 4 MIMO antenna at 5.2 GHz and 5.6 GHz. Due to the symmetrical structure of the antenna elements, this article only displays the current distribution when port 1 is excited and the other ports are terminated with a 50 Ω load. As shown in Figure 6, the current was primarily distributed along the two long edges of the outer rectangular loop and around the middle I-shaped coupling branch, as well as the inner I-shaped feed branch. In addition, the current intensity on antenna element 2 is relatively weak. It is shown that the current is largely confined within the excited antenna element, demonstrating very weak electromagnetic coupling between adjacent antenna elements and excellent isolation characteristics.

### 2.3. Parameter Scanning Analysis of the Proposed Antenna

In the design of the antenna system, the I-shaped branch of the radiating elements on the outer side of the vertical dielectric substrate and the L-shaped branch of the radiating elements on the inner side of the vertical dielectric substrate are crucial for improving the antenna resonant performance. The length and width of these elements significantly influence the antenna impedance matching, thereby affecting the antenna radiation performance. Next, we discuss the impact of two pairs of critical parameters on S parameters, the length *fl*_2_ and width *f_w_* of the short side of the L-shaped radiating patch, and the width *w*_5_ and length *l*_5_ of the I-shaped radiating patch, as shown in Figure 7 and Figure 8. By observing the variation in reflection coefficients and transmission coefficients with these parameters, it is possible to select the optimal dimensions for the structure and further analyze the working mechanism of each component in the antenna.

As shown in Figure 7a, the width *w*_5_ of the I-shaped branch considerably influences the resonant modes in the 5.0–5.5 GHz frequency band. As *w*_5_ increases, the antenna resonance point shifts to a lower frequency with an increase in the equivalent distributed capacitance at the gap of the rectangular loop. *w*_5_ has a negligible effect on the resonance in the 5.5–6.0 GHz frequency band. In addition, when *w*_5_ is greater than or equal to 1.3 mm, the central I-shaped branch connects to the lower edge of the rectangular loop, so the resonant modes are altered significantly. In this case, only one resonant mode of the antenna exists at 5.7 GHz, and the bandwidth becomes narrower. As *w*_5_ increases, the isolation deteriorates between antenna elements in the 5.0–5.7 GHz band; however, in the 5.7–5.8 GHz range, the impact of *w*_5_ on the isolation between antenna elements is negligible. As shown in Figure 7b, the length *l*_5_ of the I-shaped branch significantly affects impedance matching. When *l*_5_ is 7.3 mm, *S*_11_ is larger than −10 dB, indicating poor impedance matching. When *l*_5_ is 8.3 mm, there are two resonance modes, which increases the impedance bandwidth, and *S*_11_ is smaller than −10 dB in the 5.13–5.80 GHz operating band, indicating good impedance matching. However, when *l*_5_ is 9.3 mm, there is only one resonance point at 5.6 GHz, resulting in a narrower bandwidth. For all three values of *l*_5_, the isolation is better than 18 dB between antenna elements, demonstrating effective decoupling. Considering impedance matching, broadband characteristics, and low coupling, the optimized dimensions for the I-shaped branch of the antenna outer radiating elements are chosen to be *w*_5_ = 0.5 mm and *l*_5_ = 8.3 mm.

As shown in Figure 8a, as the width *f*_w_ of the short side of the L-shaped branch in the inner radiating elements increases, the impedance matching improves, but the decoupling performance deteriorates within the 5.13~5.80 GHz operating band. Therefore, the width of the L-shaped branch *f_w_* = 1.5 mm is selected. Figure 8b reveals that the length *fl*_2_ of the short side of the L-shaped branch considerably influences return loss and isolation. When *fl*_2_ is 1.8 mm, the antenna exhibits poor impedance matching, and the operating band is only 5.15–5.31 GHz (*S*_11_ < −10 dB). When *fl*_2_ is 2.3 mm, the antenna has two resonance points in the operating band of 5.13~5.80 GHz (*S*_11_ < −10 dB). When *fl*_2_ is 2.8 mm, the antenna resonates only at 5.8 GHz, resulting in a narrower impedance bandwidth. This indicates that exciting multiple resonant points in the same plane helps to expand the impedance bandwidth. However, as *fl*_2_ increases, the isolation between antenna elements degrades. In order to balance the operating bandwidth and isolation, the length of the L-shaped branch in the inner radiating elements is chosen to be *fl*_2_ = 2.3 mm.

## 3. Performance of Proposed 4 × 4 MIMO Wi-Fi Antenna

### 3.1. Fabrication of the Proposed Antenna 

To comprehensively demonstrate the performance of the proposed 4 × 4 MIMO antenna system, the antenna is fabricated and assembled. A prototype was fabricated as shown in Figure 9. The ground plane was soldered onto the backside of the horizontal dielectric substrate, the four vertical sides were connected to the ground plane through solder joints, and SMA connectors were added at the four ports for antenna performance testing. To validate the feasibility and practicality of the proposed antenna, the primary parameters were both simulated and measured. Due to the symmetry of the antenna layout, only the simulation and measurement results when port 1 was excited are given in this Section.

### 3.2. Simulated and Measured Results of the Proposed Antenna

HFSS is a three-dimensional electromagnetic simulation software launched by Ansys, and HFSS employs the Finite Element Method (FEM), the integral equation method (IE), and a hybrid algorithm that combines FEM and IE to provide high-frequency electromagnetic field simulation results. FEM can accurately simulate the geometry of the model, and IE is effective for open radiation and scattering problems [25,26,27]. To evaluate the performance of the designed antenna in this paper, a 3D model of the proposed 4 × 4 MIMO antenna system was established in HFSS electromagnetic simulation software using the parametric method, and the electromagnetic field distribution and relevant performance parameters of the antenna were simulated and approximated by employing FEM numerical solution, as shown in Figure 10. Additionally, the radiation performance and diversity performance of the antenna were measured by using the ZVA67 vector network analyzer and a microwave millimeter-wave antenna darkroom testing and analysis calibration system.

#### 3.2.1. S-Parameters

The simulated and measured S-parameters of the proposed 4 × 4 MIMO antenna system are shown in Figure 11. Due to the symmetry of the antenna structure, only the reflection coefficient and transmission coefficients of port 1 are presented. The S-parameters of the antenna were simulated based on HFSS, and the *S*_11_ and *S*_21_ measurement results of were obtained using a ZVA67 vector network analyzer. Some frequency shifts occurred between the simulation and measurement data, which can be attributed to manufacturing error, the soldering quality of the SMA connectors, the FR-4 substrate dielectric constant variations at different frequencies, and electromagnetic interference in the testing environment. As illustrated in Figure 11a, the measured reflection coefficient results indicate that the antenna achieves a −10 dB impedance bandwidth of approximately 15% (5.06–5.88 GHz), whereas the simulated reflection coefficient results show a −10 dB impedance bandwidth of about 13% (5.13–5.80 GHz), which effectively covers 5G partial LTE band 46 (5.15–5.95 GHz) and 5 GHz Wi-Fi 6E band (5.150–5.825 GHz). According to the measured transmission coefficient results in Figure 11b, the isolation between antenna element 1 and element 2 is better than 21 dB; the isolation between antenna element 1 and element 3 is greater than 27 dB; and for antenna element 1 and element 4, the isolation in the operating band is better than 21 dB, which indicates that the designed antenna exhibits excellent isolation characteristics between different antenna elements.

#### 3.2.2. Radiation Pattern

The radiation pattern of the antenna represents the graph of the variation of electromagnetic radiation parameters as a function of angle at a certain distance from the antenna, which provides crucial information about the radiation properties of the antenna. Assuming that the antenna system is placed at the origin of the spherical coordinate system (*r*, *θ*, *ϕ*), the normalized radiation pattern of the antenna in the far-field region at a certain distance is shown in Equation (4).
(4)$$f(\theta ,\varphi )=\frac{F(\theta ,\varphi )}{\left|F_{\max}(\theta ,\varphi )\right|},$$

Here, *F*(*θ*, *ϕ*) is the field radiation pattern function, and |*F*_max_(*θ*, *ϕ*)| is the maximum value of the field radiation pattern function. For ease of observing and plotting the radiation pattern, the two-dimensional radiation pattern is typically used in engineering. 

In this study, the antenna radiation pattern was calculated and simulated using HFSS by configuring the far-field environment. Additionally, the actual radiation pattern of the antenna was measured in the microwave millimeter-wave antenna darkroom, as shown in Figure 12.

Due to the symmetry of the antenna structure, only the simulation and measurement results of antenna element 1 at the frequencies of 5.2 GHz and 5.8 GHz are shown. The two-dimensional (2D) radiation patterns in the xoy plane (Theta = 0°), xoz plane (Phi = 0°), and yoz plane (Phi = 90°) are depicted in Figure 13. By comparing the simulation and measurement results, it is evident that the trends of the simulated and measured curves are consistent; however, the sharps of the radiation patterns exhibit slight variations due to the influence of resonant modes on the operating frequency. The differences between the simulated and measured 2D radiation patterns are attributed to manufacturing and measurement errors. Although the radiation pattern of the antenna element is not strictly omnidirectional in the operating bandwidth, it exhibits no nulls in the observation plane and approximates omnidirectional radiation characteristics, making it suitable for application in 5G mobile terminal devices.

#### 3.2.3. Radiation Efficiency and Peak Gain

When an antenna transmits and receives electromagnetic waves, not all of the electromagnetic energy is effectively radiated into space, as there is a certain amount of energy loss. The efficiency of the antenna is used to evaluate the radiation performance of the antenna, which is defined as the ratio of the radiated power to the total input power, as shown in Equation (5).
(5)$$\eta =\frac{P_{rad}}{P_{in}}=\frac{P_{in}-P_{loss}}{P_{in}}=1-\frac{P_{loss}}{P_{in}},$$

Here, *P_rad_* denotes the radiated power of the antenna, *P_in_* denotes the total input power of the antenna, and *P_loss_* denotes the loss power of the antenna. In MIMO antenna systems, when one antenna element is excited, adjacent antennas act as loads. Due to the mutual coupling effects, a portion of the input power is absorbed by these loads, resulting in a reduction in the antenna radiation efficiency. Typically, the efficiency of the MIMO antenna in mobile terminals is required to be higher than 40%. 

The gain of an antenna is a parameter that measures the strength of its ability to radiate in a specific direction, taking into account the power loss, and reflects its effectiveness in capturing or transmitting signals of the antenna. The gain *G* of the antenna can be calculated by multiplying the directivity factor *D* with the efficiency *η* of the antenna, as shown in Equation (6).
(6)G=η×D,

The radiation efficiency and peak gain of the proposed antenna were simulated by HFSS, and the test method of that was the same as the radiation pattern, as shown in Figure 14. The simulated and measured radiation efficiencies of the proposed antenna are shown in Figure 14a. The simulated and measured radiation efficiencies of antenna element 1 are 51.2–89.9% and 51.8–92.3%, respectively, in the operational bandwidth of 5.06–5.88 GHz, which can meet the requirement for a total antenna efficiency greater than 50% for mobile terminals. The simulated and measured peak gains of the antenna are shown in Figure 14b. When port 1 is excited, the simulated peak gain of the antenna ranges from 2.8 dBi to 4.5 dBi, while the measured peak gain ranges from 2.4 dBi to 5.3 dBi in the operational bandwidth. At the 5.6 GHz frequency point, the measured maximum gain of the antenna reaches 5.3 dBi, satisfying the general gain design requirements of 2 dBi–5 dBi for Wi-Fi antennas. The discrepancies between the measured and simulated results are primarily attributed to the physical assembly and soldering of the antenna components.

#### 3.2.4. Diversity Performance

The envelope correlation coefficients (ECCs) can be used to evaluate the diversity performance of an antenna system in the radiating space, which can indicate the spatial correlation level of the radiating waves of two antennas and characterize the overall efficiency and reliability of the antenna. The ECCs are deduced by the simulated and measured far-field radiation patterns or approximated using Equation (7), as indicated in reference [28].
(7)$$ECC(i,j)=\frac{\left|S_{ii}^{*}S_{ij}+S_{ji}^{*}S_{jj}\right|}{\left(1-({\left|S_{ii}\right|}^{2}+{\left|S_{ji}\right|}^{2}))(1-({\left|S_{jj}\right|}^{2}+{\left|S_{ij}\right|}^{2})\right)},$$

Here, *S_ii_* and *S_jj_* denote the reflection coefficients of antenna elements *i* and *j*, respectively; *S_ij_* and *S_ji_* denote the transmission parameters between antenna elements *i* and *j*; and $S_{ii}^{*}$ and $S_{ji}^{*}$ denote the conjugate complexes of *S_ii_* and *S_jj_*. The measured and simulated ECCs of the proposed 4 × 4 MIMO antenna system are shown in Figure 15. In practical MIMO antenna systems, a lower ECC value indicates a smaller correlation between the antennas and a stronger spatial diversity ability, which helps to mitigate multipath fading effects and enhances the resistance against interference in the wireless communication channel. It can be observed that the ECCs between antenna elements 1 and 2 are consistently below 0.05 (the acceptable value of the ECC is less than 0.5) in the desired band. This indicates that the proposed 4 × 4 MIMO antenna system exhibits low spatial correlation and excellent diversity performance.

#### 3.2.5. Diversity Gain

Diversity gain (DG) is another metric which was used to evaluate the diversity performance of MIMO antenna systems. DG is calculated based on the envelope correlation coefficient (ECC), as detailed in Equation (8) in reference [29]
(8)$$DG(i,j)=10\sqrt{1-ECC{\left(i,j\right)}^{2}},$$

Typically, the DG of MIMO antennas in mobile terminals is required to be close to 10 dB, which indicates that the MIMO antenna system has good interference resistance in the multipath fading environment. Due to the symmetrical arrangement of the antenna structure, only the diversity gain (DG) between element 1 and element 2 of the proposed antenna is calculated in this paper, as shown in Figure 16.

It can be observed that in the operating band of 5.06~5.88 GHz, the value of DG is better than 9.98 dB, which indicates that the MIMO antenna system has excellent spatial diversity capability. This demonstrates that the antenna can effectively utilize the multipath propagation to mitigate the signal fading and enhance the reliability of communication.

To highlight the advantages of the proposed 4 × 4 MIMO antenna, it is compared to the conventional 4 × 4 MIMO mobile terminal antennas reported in references [30,31,32,33] as listed in Table 2. 

## 4. Conclusions

This article proposes a wideband, low coupling, and miniaturized 4 × 4 MIMO antenna design for WiFi. The antenna consists of four antenna elements with multi-branch structures, and the dimensions of each antenna element are 15.2 × 3.5 × 0.8 mm^3^. The wideband characteristics are achieved through the multimode resonance principle, and orthogonally arranged antenna elements ensure that the main radiation directions of the adjacent antenna elements point to different regions, which achieves the performance of low coupling among the antenna elements without any additional decoupling structure. The simulation results show that in the 5.13–5.80 GHz band (*S*_11_ < −10 dB), the isolation between antenna elements is greater than 20 dB, and the radiation efficiencies are 51.2–89.9%, respectively; the peak gain reaches 4.5 dBi and the ECCs are below 0.04. The measurement results indicate that in the 5.06–5.88 GHz band (*S*_11_ < −10 dB), the antenna isolation between the elements is greater than 21 dB, the radiation efficiencies are 51.8–92.3%, the maximum gain reaches 5.3 dBi, and the ECCs are less than 0.05. The results indicate that the proposed 4 × 4 MIMO antenna system effectively balances the requirements of antenna size, bandwidth, and isolation, exhibiting low coupling, wide bandwidth, high efficiency, a compact size, and a simple structure. These superior performance characteristics highlight its significant application potential in 5G mobile terminal devices.

## Figures and Tables

**Figure 1 micromachines-15-00850-f001:**
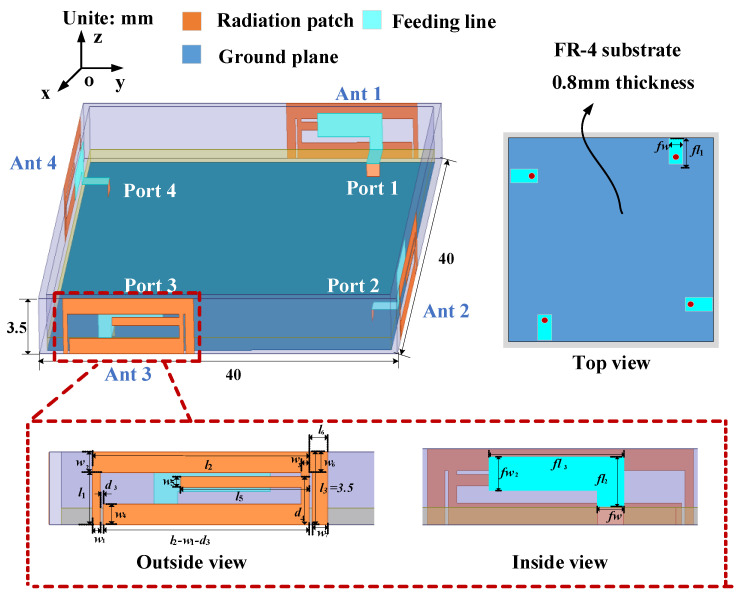
Geometry of the proposed 4 × 4 MIMO (multiple-input multiple-output) antenna system.

**Figure 2 micromachines-15-00850-f002:**
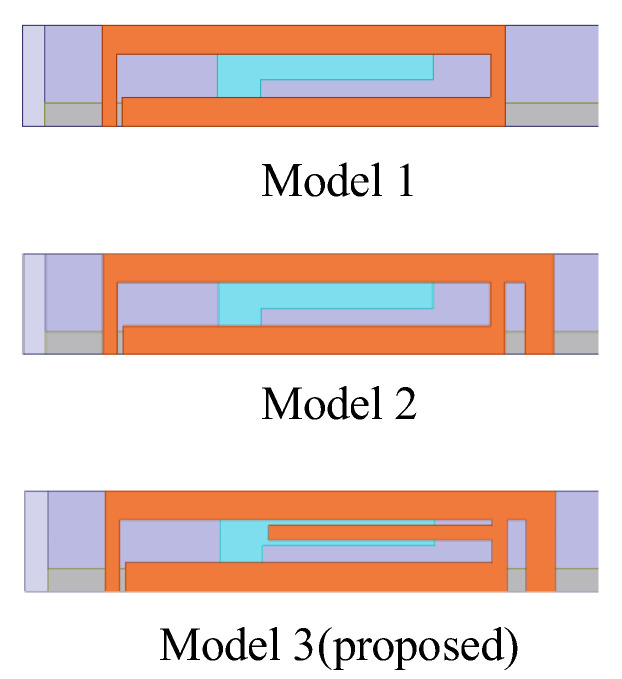
Evolution of the proposed antenna element structure.

**Figure 3 micromachines-15-00850-f003:**
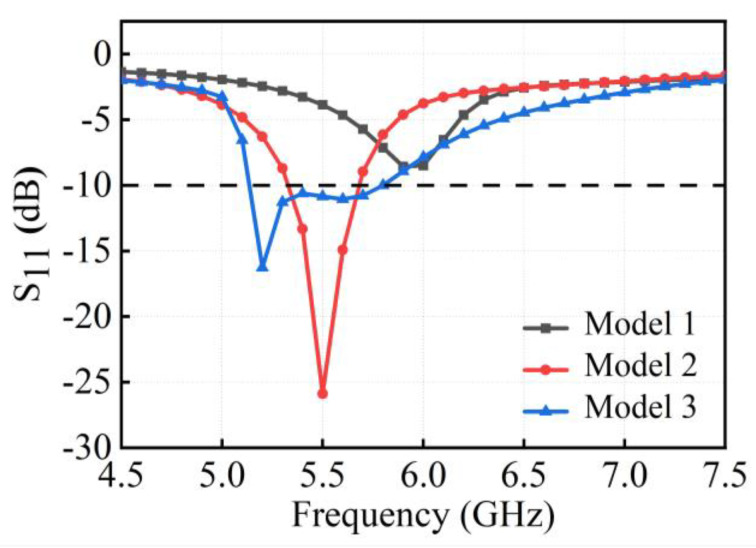
Variations of reflection coefficients for three evolutionary models.

**Figure 4 micromachines-15-00850-f004:**
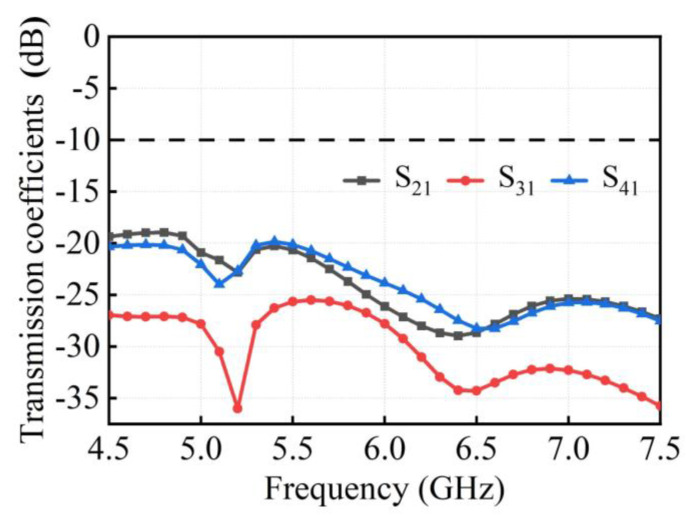
Simulated transmission coefficients of the proposed 4 × 4 MIMO antenna.

**Figure 5 micromachines-15-00850-f005:**
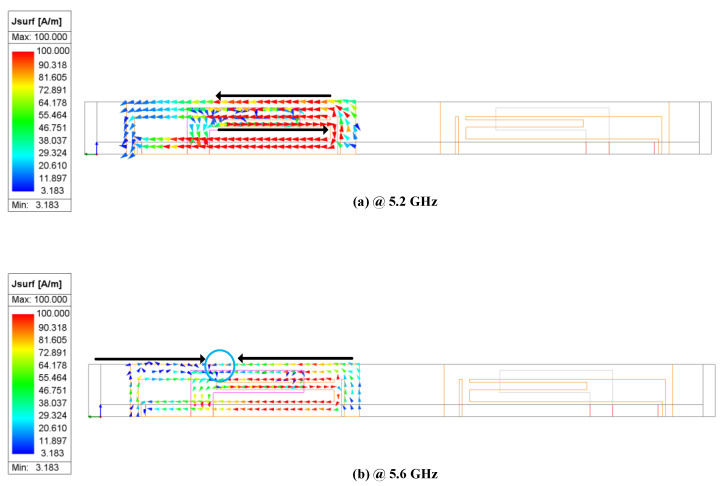
Simulated surface current models of the proposed antenna element 1. (**a**) 5.2 GHz; (**b**) 5.6 GHz.

**Figure 6 micromachines-15-00850-f006:**
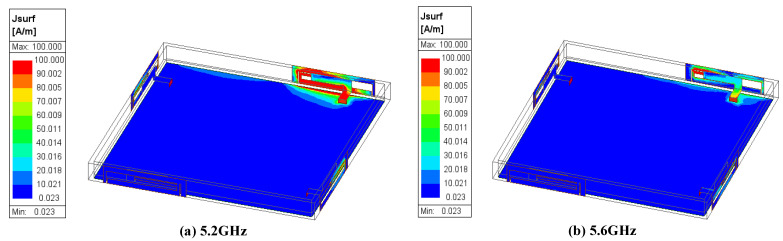
Simulated surface current distributions of the proposed antenna when port 1 is excited. (**a**) 5.2 GHz; (**b**) 5.6 GHz.

**Figure 7 micromachines-15-00850-f007:**
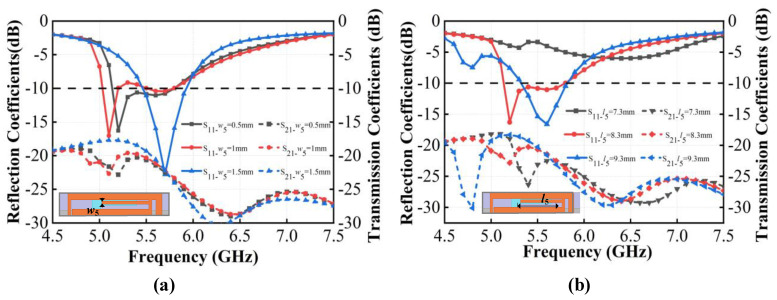
Simulated S parameters of the proposed antenna with different values of the I-shaped branch in the outer radiating elements. (**a**) *w*_5_; (**b**) *l*_5_.

**Figure 8 micromachines-15-00850-f008:**
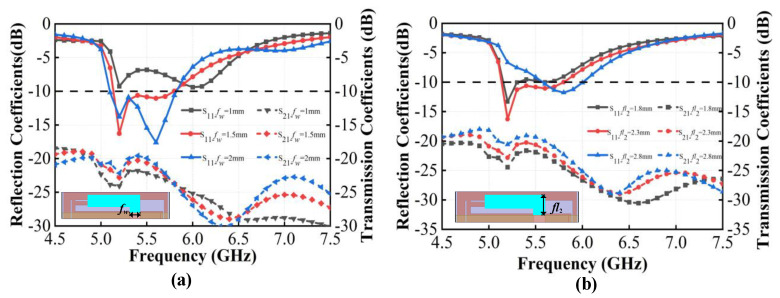
Simulated S parameters of the proposed antenna with different values of the L-shaped branch in the inner radiating elements. (**a**) *f_w_*; (**b**) *fl*_2_.

**Figure 9 micromachines-15-00850-f009:**
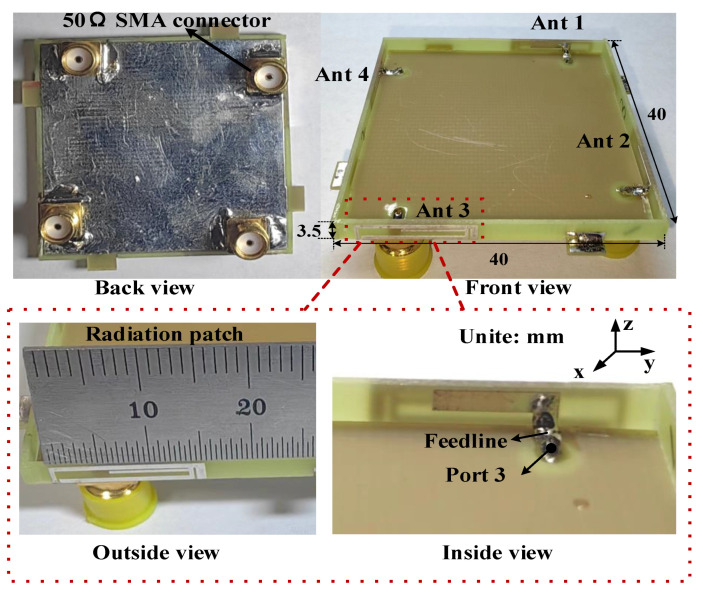
Photograph of the prototype of the proposed 4 × 4 MIMO antenna system.

**Figure 10 micromachines-15-00850-f010:**
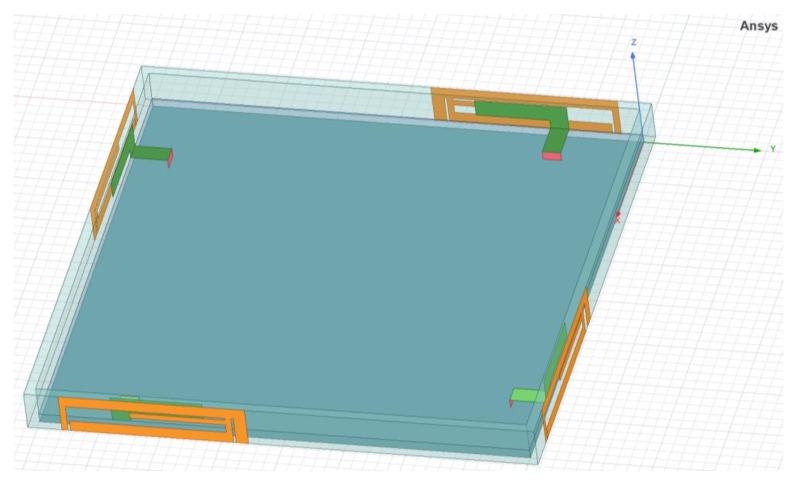
Simulation model of the proposed 4 × 4 MIMO antenna system.

**Figure 11 micromachines-15-00850-f011:**
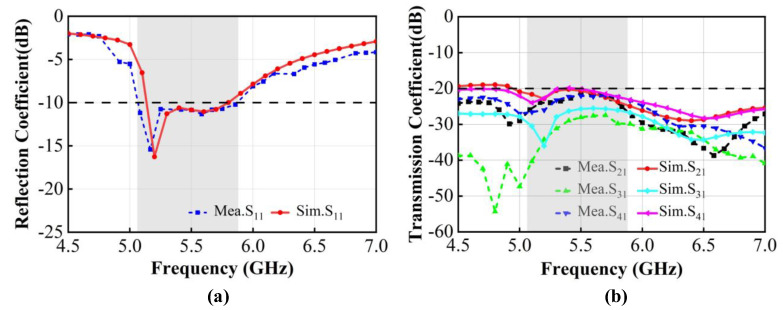
Simulated and measured S-parameters of the proposed 4 × 4 MIMO antenna system. (**a**) Reflection coefficients *S*_11_. (**b**) Transmission coefficients.

**Figure 12 micromachines-15-00850-f012:**
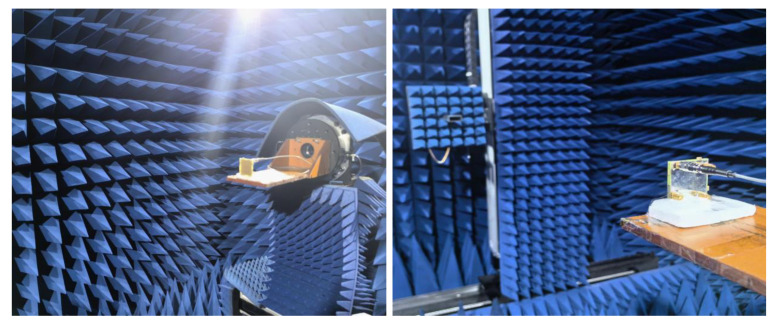
Antenna radiation performance measurement scenario.

**Figure 13 micromachines-15-00850-f013:**
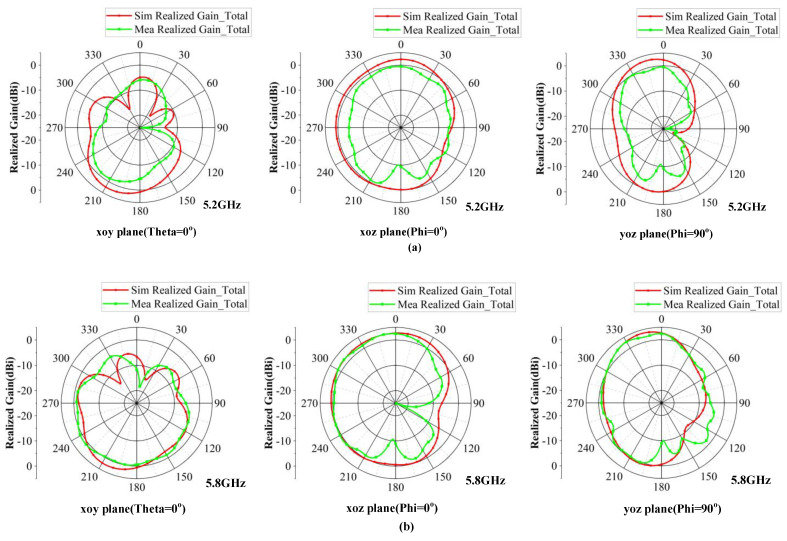
Simulation and measurement 2D radiation patterns of the proposed antenna at the frequencies of 5.2 GHz and 5.8 GHz (angle unit: degree). (**a**) 5.2 GHz; (**b**) 5.8 GHz.

**Figure 14 micromachines-15-00850-f014:**
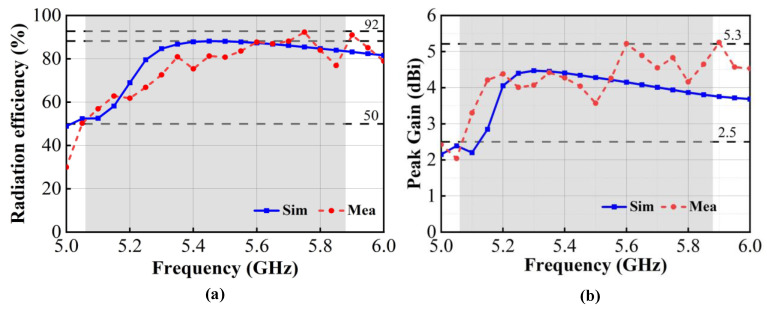
Simulated and measured radiation efficiency and peak gain of the proposed antenna. (**a**) Radiation efficiency; (**b**) peak gain.

**Figure 15 micromachines-15-00850-f015:**
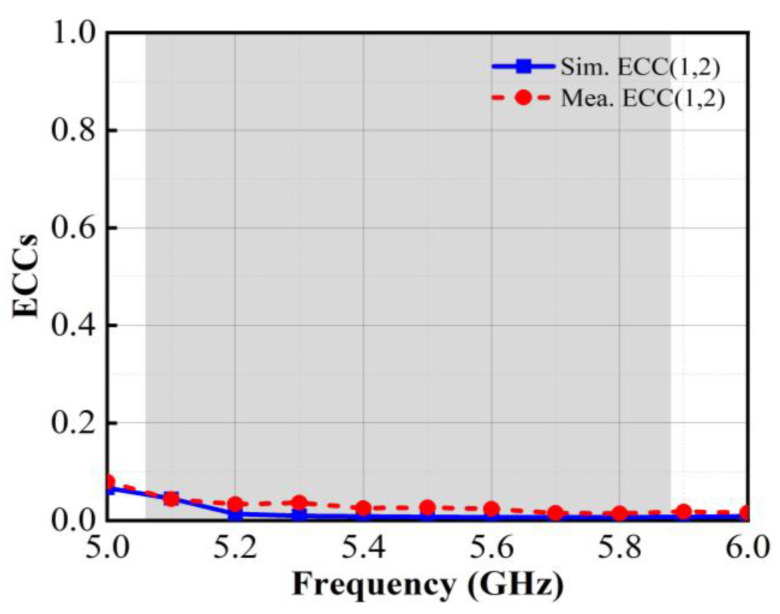
Simulated and measured ECCs of the proposed antenna.

**Figure 16 micromachines-15-00850-f016:**
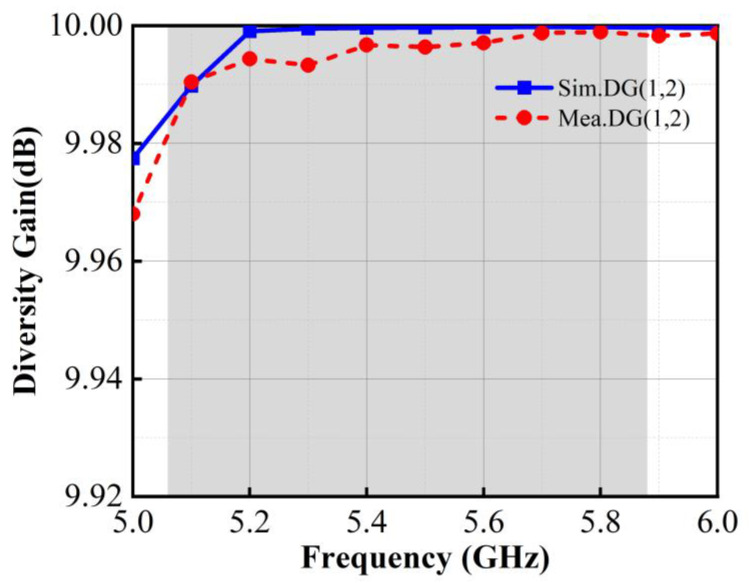
Calculated DG of the proposed antenna.

**Table 1 micromachines-15-00850-t001:** The detailed dimensions of the optimized antenna geometry.

**parameter**	*l* _1_	*w* _1_	*l* _2_	*w* _2_	*w* _3_	*d* _3_	*w* _4_	*d* _4_	*l* _5_
**value/mm**	2.5	0.5	14	1	0.5	0.2	1	2.3	8.3
**parameter**	*w* _5_	*l* _6_	*w* _6_	*w* _7_	*fl* _1_	*f_w_*	*fl* _2_	*fl* _3_	*fw* _2_
**value/mm**	0.5	1.2	1	1	3	1.5	2.3	7.5	1.5

**Table 2 micromachines-15-00850-t002:** Comparison of antenna performance of 4 × 4 MIMO antenna of mobile terminals [30,31,32,33].

Ref.	Antenna Element Size/mm^2^	Decoupling Mechanisim	Operating Band /GHz	Isolation /dB	Efficiency /%
[30]	14.9 × 7	-	3.4~3.6/4.8~5.0	>16.5	74~85/70~82
[31]	15.5 × 10.5	DGS	5.61~5.9	>18.2	>76
[32]	20 × 6.2	PEs	3.4–3.6	>17	—
[33]	13.5 × 13.5	PE	0.85–0.94/2–12	>20	—
Proposed	15.2 × 3.5	-	5.06–5.88	>21	5–92.3%

## Data Availability

The original contributions presented in the study are included in the article, further inquiries can be directed to the corresponding authors.

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
