# Peer review of "Design of Miniaturized and Wideband Four-Port MIMO Antenna Pair for WiFi"

_micromachines, 2024, doi:10.3390/mi15070850_

Round 1

Reviewer 1 Report

Comments and Suggestions for Authors

In this study, a miniaturized broadband four-port MIMO antenna for Wi-Fi applications is proposed by combining the radiation mechanism of the antenna and the internal decoupling technique of radiation direction diversity. Four identical antenna elements are orthogonally arranged to ensure independent signal radiation. This configuration avoids adding complexity in the antenna structure and enhances spatial efficiency. This design meets the requirements for miniaturization, wide bandwidth, and high isolation.  The simulation and test results confirm that the radiation performance of the proposed 4×4 MIMO Wi-Fi antenna satisfies the operational demands of 5G mobile devices.

Some shortcomings and omissions of the paper are the following:

1. The paper presents only model and test numerical results. The description of simulation methods, experimental technique and test equipment is absent. Obviously, these topics must have separate sections in the paper.

2. (Line 143): How are the transmission and reflection coefficients simulated?

3. (Line 248): How are the S-parameters of the proposed 4×4 MIMO antenna system simulated and measured?

4. (Line 272-275): How is the radiation pattern of the antenna simulated and measured?

5. (Line 292): How are the radiation efficiency and peak gain of the proposed antenna simulated and measured?

6. What does (*) mean in Expression (1)?

7. Formulae of other authors must be accompanied by corresponding references. For example (2).

8. References into text must be pointed in square brackets. For example: 2 and 6 (Line 55).

9. List of References breaks the MDPI standards for references and all papers must be accompanied by doi.

Comments on the Quality of English Language

There are numerous mistakes in English language. The paper should be checked by native English speaker. For example:

(Line 105): “…through via holes.”

(Line 108): “Geometry of the geometry...”

(Line 112): “…the paper has been chosen to analyze the evolution...” (What is the paper?)

(Line 121): “The evolution of the proposed antenna element structure as shown...”

(Line 201): “…the antenna exists only one resonant mode...”

(Line 221-222): “…as the width fw of the short side of the L-shaped in the inner radiating elements, the impedance matching improves...”

(Line 322): “…system re shown...”

Author Response

  1. Summary

Thank you very much for taking the time to review this manuscript. Please find the detailed responses below and the corresponding revisions in track changes in the re-submitted files.

  1. Point-by-point response to Comments and Suggestions for Authors

Comments 1: The paper presents only model and test numerical results. The description of simulation methods, experimental technique and test equipment is absent. Obviously, these topics must have separate sections in the paper.

Response 1: The simulation methods and experimental test equipment are described in the final paragraph of the abstract section. Additionally, the specific measurement and simulation methods are specified in conjunction with the antenna performance parameters in Chapter 3 section.

Comments 2: (Line 143): How are the transmission and reflection coefficients simulated?

Response 2: The transmission and reflection coefficients of the antenna are simulated by HFSS, and has been added in Line 148.

Comments 3: How are the S-parameters of the proposed 4×4 MIMO antenna system simulated and measured?

Response 3: The S-parameters of the antenna are simulated based by HFSS, and the S11 and S21 measurement results of are obtained using an ZVA67 vector network analyzer, and this section has been added in Line 255-257.

Comments 4: (Line 272-275): How is the radiation pattern of the antenna simulated and measured?

Response 4: The antenna radiation pattern are simulated using HFSS by configuring the far-field environment. Additionally, the actual radiation pattern of the antenna was measured in the microwave millimeter-wave antenna darkroom, and this section has been added in Line 280-282.

Comments 5: (Line 292): How are the radiation efficiency and peak gain of the proposed antenna simulated and measured?

Response 5: The radiation efficiency and peak gain of the proposed antenna are simulated by HFSS, and the test method of that is the same as the radiation pattern in Line 300-302.

Comments 6: What does (*) mean in Expression (1)?

Response 6: The * in Expression (1) is specifically explained in the paper.

Comments 7: Formulae of other authors must be accompanied by corresponding references. For example (2).

Response 7: Reference 26 has been added in Equation (2), and the subsequent references have been renumbered accordingly.

Comments 8: References into text must be pointed in square brackets. For example: 2 and 6 (Line 55).

Response 8: All references in the text have been pointed in square brackets.

Comments 9: List of References breaks the MDPI standards for references and all papers must be accompanied by doi.

Response 9: Except for references 12 and 28 which do not have doi, all other list of references have been accompanied by doi.

  1. Response to Comments on the Quality of English Language

Point 1: (Line 105): “…through via holes.”

Response 1: “…through via holes.” is modified to “via holes”.

Point 2: (Line 108): “Geometry of the geometry...”

Response 2: “Geometry of the geometry...”is modified to “Geometry of the proposed 4×4 MIMO antenna system.”

Point 3: (Line 112): “…the paper has been chosen to analyze the evolution...” (What is the paper?)

Response 3: “…the paper has been chosen to analyze…”is modified to “…the section focuses on analyzing…”

Point 4: (Line 121): “The evolution of the proposed antenna element structure as shown...”

Response 4: This has been modified to “The evolution process of the proposed antenna element structure is shown in Figure 2, and the variation of reflection coefficient is illustrated in Figure 3. ”

Point 5: (Line 201): “…the antenna exists only one resonant mode...”

Response 5: This has been modified to“And when w5 is greater than or equal to 1.3 mm, the central I-shaped branch connects to the lower edge of the rectangular loop, so the resonant modes are altered significantly. In this case, only one resonant mode of the antenna exists at 5.7 GHz, and the bandwidth becomes narrower.”

Point 6: (Line 221-222): “…as the width fw of the short side of the L-shaped in the inner radiating elements, the impedance matching improves...”

Response 6: This has been modified to “As shown in Figure 8(a), as the width fw of the short side of the L-shaped in the inner radiating elements increases, the impedance matching improves but the decoupling performance deteriorates within the 5.13~5.80 GHz operating band.”

Point 7: (Line 322): “…system re shown...”

Response 7: This has been modified to “The measured and simulated ECCs of the proposed 4×4 MIMO antenna system are shown in Figure 14.”

Reviewer 2 Report

Comments and Suggestions for Authors

The manuscript reports the engineering design of a wideband multiple antenna for 5G/6G/WiFi devices that fulfill in some important respects the criteria to be used in those systems.

1) As strong points, the authors detail clearly the mechanical and electrical design of the antenna, so that the fabrication process is quite straightforward, and also simulate comprehensively and extensively the electromagnetic properties. This numerical results confirm that the antenna satisfies the design requirements.

2) The simulated measurements on the antenna can be presented with figures with increased resolution, to make them easier to follow (e.g., in Figs. 10 and 13 onwards. 

3) Please, make the bibliographic entries more uniform in their presentation. In the current manuscript, these are mixed in a somewhat confusing way.

Author Response

  1. Summary

Thank you very much for taking the time to review this manuscript. Please find the detailed responses below and the corresponding revisions in track changes in the re-submitted files.

  1. Point-by-point response to Comments and Suggestions for Authors

Comments 1: As strong points, the authors detail clearly the mechanical and electrical design of the antenna, so that the fabrication process is quite straightforward, and also simulate comprehensively and extensively the electromagnetic properties. This numerical results confirm that the antenna satisfies the design requirements.

Response 1: Thank you to the reviewer for your comments on my paper.

Comments 2: The simulated measurements on the antenna can be presented with figures with increased resolution, to make them easier to follow (e.g., in Figs. 10 and 13 onwards). 

Response 2: I have replaced the figures of the simulated measurements on the antenna with increased resolution(e.g., in Figs. 10 and 13 onwards).

Comments 3: Please, make the bibliographic entries more uniform in their presentation. In the current manuscript, these are mixed in a somewhat confusing way.

Response 3:  In the manuscript, I have revised all the bibliographic entries to make them uniform.

Round 2

Reviewer 1 Report

Comments and Suggestions for Authors

The authors have introduced certain revision, allowing one to improve this paper. However, the following significant questions remain:

1. Again, the paper presents only model and test numerical results. The description of simulation methods, experimental technique and test equipment is absent. The phrase to describe simulation method in Lines 84-85 is unsatisfactory: “The characteristics and application scenarios of the MIMO antenna were simulated and analyzed by HFSS electromagnetic simulation software.”

Abbreviation HFSS must be deciphered. Who is developer of this software? What is this software? What are the main constitutive equations? What is the numerical method used (analytical approach, finite-element method, finite-difference method or another?).

The same questions there are for the phrase, describing experimental modeling in Lines 88-91:

“Meanwhile, the S-parameters and radiation performance of the antenna were measured using ZVA67 vector network analyzer and a microwave millimeter-wave antenna darkroom testing and analysis calibration system.”

Who is developer of the equipment? How were processed experimental data (what are mathematical formulae used?)

Obviously, these topics must have separate sections in the paper.

2. List of References breaks the MDPI standards for references. The right example for references is the following:

SchattlingJochumTheatoMulti-stimuli responsive polymers – the all-in-one talents. Polymer Chemistry 2014, 5, 25-36, https://doi.org/10.1039/C3PY00880K

Author Response

Comments 1:  Again, the paper presents only model and test numerical results. The description of simulation methods, experimental technique and test equipment is absent. The phrase to describe simulation method in Lines 84-85 is unsatisfactory: “The characteristics and application scenarios of the MIMO antenna were simulated and analyzed by HFSS electromagnetic simulation software.”

Abbreviation HFSS must be deciphered. Who is developer of this software? What is this software? What are the main constitutive equations? What is the numerical method used (analytical approach, finite-element method, finite-difference method or another?).

The same questions there are for the phrase, describing experimental modeling in Lines 88-91:

“Meanwhile, the S-parameters and radiation performance of the antenna were measured using ZVA67 vector network analyzer and a microwave millimeter-wave antenna darkroom testing and analysis calibration system.”

Who is developer of the equipment? How were processed experimental data (what are mathematical formulae used?)

Obviously, these topics must have separate sections in the paper.

Response 1: The description of simulation methods, experimental technique and test equipment are added additional clarifications in Line 85, Lines 126-156, 277-284, 310-318, 342-358, 365-370. The equations have been renumbered accordingly in Line 386-388 and 406-407. HFSS is a three-dimensional electromagnetic simulation software launched by Ansys, and HFSS employs the Finite Element Method (FEM) and the integral equation method to provide high-frequency electromagnetic field simulation results. Additionally, the radiation performance and diversity performance of the antenna are measured by using the ZVA67 vector network analyzer and a microwave millimeter-wave antenna darkroom testing and analysis calibration system in Interconnection and Sensing Microelectronics Laboratory, Tianjin University, China.

Comments 2:  List of References breaks the MDPI standards for references. The right example for references is the following:

Schattling,  P.; Jochum,  F. D.; Theato, P. Multi-stimuli responsive polymers – the all-in-one talents. Polymer Chemistry 2014, 5, 25-36, https://doi.org/10.1039/C3PY00880K

Response 2: The list of references has been modified in accordance with MDPI standards for references in Lines 457-579.

Round 3

Reviewer 1 Report

Comments and Suggestions for Authors

The authors have made necessary revisions to the manuscript. However, HFSS electromagnetic simulation software must be supplemented by corresponding reference. After that, the paper could be accepted to publication.

Author Response

References 25-27 related to the HFSS electromagnetic simulation software have been added at line 281, and the content from lines 277-286 has been revised. Additionally, simulation model of the proposed 4×4 MIMO antenna system as depicted in Figure 10 has been inserted below line 289. All modifications in this instance are highlighted in green.
